# CO_2_ Adsorption Study of Potassium-Based Activation of Carbon Spheres

**DOI:** 10.3390/molecules27175379

**Published:** 2022-08-23

**Authors:** Iwona Pełech, Piotr Staciwa, Daniel Sibera, Robert Pełech, Konrad S. Sobczuk, Gulsen Yagmur Kayalar, Urszula Narkiewicz, Robert Cormia

**Affiliations:** 1Department of Chemical and Environment Engineering, Faculty of Chemical Technology and Engineering, West Pomeranian University of Technology in Szczecin, Pułaskiego 10, 70-322 Szczecin, Poland; 2Faculty of Civil and Environmental Engineering, West Pomeranian University of Technology in Szczecin, al. Piastów 50a, 70-311 Szczecin, Poland; 3Department of Chemical Organic Technology and Polymeric Materials, Faculty of Chemical Technology and Engineering, West Pomeranian University of Technology in Szczecin, Pułaskiego 10, 70-322 Szczecin, Poland; 4Department of Chemical Engineering, Faculty of Engineering, Eskişehir Technical University, 26555 Eskişehir, Turkey; 5Chemistry Faculty, Chemistry Department, Foothill College, 12345 El Monte Road, Los Altos Hills, CA 94022, USA

**Keywords:** carbon spheres, activation, potassium hydroxide, carbon dioxide, adsorption

## Abstract

The adsorption properties of microporous spherical carbon materials obtained from the resorcinol-formaldehyde resin, treated in a solvothermal reactor heated with microwaves and then subjected to carbonization, are presented. The potassium-based activation of carbon spheres was carried out in two ways: solution-based and solid-based methods. The effect of various factors, such as chemical agent selection, chemical activating agent content, and the temperature or time of activation, was investigated. The influence of microwave treatment on the adsorption properties was also investigated and described. The adsorption performance of carbon spheres was evaluated in detail by examining CO_2_ adsorption from the gas phase.

## 1. Introduction

The substantial and continuous increase in environmental pollution levels over recent years, particularly in developed and developing countries, has inspired research to solve this universal problem. The responsibility for the growing air pollution has primarily been assigned to large-scale human activities, especially the reliance on fossil fuels. Industrial and energy production often lacks the necessary precautions to completely rule out the possibility of their contributing to environmental pollution, but developed countries are currently trying to find a solution to this problem, e.g., using systems such as Carbon Capture and Storage (CCS) [1,2]. An example of CCS confirms that the adsorption of greenhouse gas pollutants is generally considered the best method of adsorbing gases such as CO_2_ due to its low cost, high efficiency, and easy regeneration. Many new materials have recently been studied to remove carbon dioxide. Notable examples include the use of highly porous materials such as microporous organic polymers (MOPs) [3,4] or metal–organic frameworks (MOFs) [5,6]. The continuation of trends regarding porosity as a key property of the desired adsorption material, coupled with the need to use more sustainable and widespread materials, has led to an extensive investigation of different types of carbon materials. In recent decades, research on carbon materials in terms of their use as a potential adsorbent of various impurities has included activated carbons (ACs) [7,8], metal–carbon composites [9,10], bio-waste-derived carbons [11,12], and nitrogen-doped carbons [13,14]. The increased interest in describing activated carbon materials is most importantly associated with their physical properties, such as a large porous volume, well-defined pore size distribution, and as a result, a high specific surface area.

Carbon spheres are interesting materials, which attract a great deal of attention due to their important physical properties, such as their mechanical strength, low ash content, purity, good fluidity, wear resistance and durability, high surface area and micropore volume, and tunable pore size distribution [15,16,17,18,19]. Additionally, the low cost of processing, chemical stability, and the possibility of modification with heteroatoms [20] also significantly increases the widespread research impact of carbon materials at present. A large porous volume, well-defined pore size distribution and a high specific surface area show that they are materials with good adsorption properties, which can be developed as a result of the activation processes [21,22,23,24].

The term “activation”, regarding carbon spheres, usually means that carbon reestablishes and develops a high active surface area after the carbonization process. The increased porosity and structural defects of the end-product are also good predictors of final reactivity. Carbon or carbonized materials can be activated via chemical or/and physical processes [16,25,26,27,28,29,30,31,32,33]. Additionally, to further improve the performance of certain carbon materials, and find an optimal path for their preparation, a combination of both pathways (chemical and physical) can be used [34,35]. Physical activation occurs when carbonized materials are treated at approximately 700–900 °C in the presence of oxidizing gases, mainly steam or carbon dioxide [16,25,26]. However, the adsorption capacity and surface area of the carbonized materials are limited due to the recrystallization processes and the presence of tarry products in the structure. The use of oxidizing gases aims to remove tarry products, create new pores and increase the diameter of already existing pores at high temperatures due to the occurrence of localized weight loss [19]. Chemical activation involves impregnating raw materials with certain chemical agents and then thermally decomposing them in a single step, which allows for an increased BET surface area [17]. Activated carbon spheres can be obtained by mixing the carbon spheres with a chemical agent, such as ZnCl_2_ [27], H_3_PO_4_ [28], KOH [29], K_2_CO_3_ [30], K_2_C_2_O_4_ [31,32], NaOH [33]. These substances act both as dehydrating agents and as oxidants, ensuring that carbonization and activation can take place simultaneously. The obtained mixture is then dried and subjected to heat treatment under an inert atmosphere at temperatures between 400 °C and 700 °C [36].

Several reports have been presented on the effectiveness of KOH activated carbon spheres in the adsorption of carbon dioxide from air or fumes. Xuan et al. [22] prepared porous carbon spheres by the direct carbonization of resorcinol-formaldehyde resin spheres with a potassium salt and examined the CO_2_ adsorption potential of the produced material. The prepared materials exhibited desired properties such as a high CO_2_ capture performance of up to 4.83 mmol/g, good CO_2_ selectivity against N_2_, and stable recyclability. Ludwinowicz and Jaroniec [31] studied the influence of different activating agents on the porosity of carbon spheres and their CO_2_ adsorption properties. It was shown that KOH activation yielded the highest volume of micropores and small micropores; in addition, the CO_2_ uptake for the KOH-activated sample was the highest. Wang et al. [37] used hemp stem hemicellulose as a precursor to prepare well-shaped carbon spheres with a large surface area of up to 3062 m^2^/g using an improved low-temperature hydrothermal carbonization method and KOH activation. They indicated that the plentiful micropores lead to excellent CO_2_ and CH_4_ adsorption capacities at ambient pressure and 0 °C. Dassanayake and Jaroniec [38] obtained a high inherent microporosity of polypyrrole-derived carbon spheres, which was further optimized by controlled KOH activation with the intention to achieve high CO_2_ adsorption capacity. The high ultramicropore volumes of these spheres resulted in superior CO_2_ uptakes under ambient pressure, reaching 7.73 mmol/g and 5.42 mmol/g at 0 °C and 25 °C, respectively. These results suggest satisfying prospects for their future application.

This work aimed to characterize the adsorption capacity of microporous spherical carbon materials obtained by the carbonization of resorcinol-formaldehyde resin with the pore structure improved by the activation process. The influence of chemical agent selection, chemical activating agent content, activation temperature and time, as well as microwave treatment, on the textural properties and morphology of the obtained carbon materials was determined. Reactions carried out in the microwave-assisted solvothermal reactor are very fast (min) compared with those conducted in an autoclave (h). The use of microwaves results in a very low temperature gradient in the reactor volume, which can be neglected. Microwaves affect the polar solvents; then, in the whole reaction volume, nucleuses are rapidly created, enabling small particles with a low size distribution to be obtained. The adsorption performance of carbon spheres was evaluated in detail by examining CO_2_ capture.

## 2. Results and Discussion

### 2.1. Morphology of the Obtained Materials

Scanning Electron Microscopy was used to investigate the changes in the morphology of carbon spheres after treatment in different reagents. The detailed characteristics of carbon spheres obtained under different experimental conditions have been presented in our previous works [39,40]. The selected SEM images of carbon spheres activated using the solid-based method with KOH content of 10, and 50 wt.% and solution-based method with KOH content of 10, and 70 wt.% are presented in Figure 1.

It was found that the activation with KOH content of 10 wt.% (Figure 1a), or even 50 wt.% (Figure 1b), did not destroy the spherical shape of carbon. The same effect was observed for the materials activated using a solution-based method with KOH content of 10 (Figure 1c), 20, 50, or even 70 wt.% (Figure 1d). However, in the case of the samples that were activated using a solid- or solution-based method with KOH content of 50 wt.% or higher, more non-spherical carbon inclusions were observed (Figure 1b,d).

### 2.2. Surface Area and Pore Size Distribution Studies

The specific surface area (S_BET_), total pore volume (TPV), volume of micropores (V_m_ < 2 nm), and volume of mesopores (V_meso_) values of the tested materials, determined on the basis of low-temperature (−196 °C) nitrogen adsorption isotherms, as presented in Figure 2, Figure 3, Figure 4 and Figure 5, are listed in Table 1. From CO_2_ adsorption isotherms at 0 °C (presented in the Section 2.3.), pore size distribution (PSD) and the volume of ultramicropores V_s_ with dimensions smaller than 1.0 nm (<1 nm) were determined and calculated by integrating the pore volume distribution function using the NLDFT method.

Comparing the samples activated with different potassium hydroxide content and using a solid-based (K_10S_700_05, K_20S_700_05, K_50S_700_05) (Table 1, Figure 2) or solution-based method (K_10L_700_05, K_20L_700_05, K_50L_700_05, K_70L_700_05) (Table 1, Figure 3) with the reference material (RF), it was found that activation process caused an increase in the specific surface area and total pore volume in almost all cases. The materials activated with 10 wt.% solid KOH adsorbed the smallest amount of nitrogen, which was characterized by the lowest values of the specific surface area of 480 m^2^/g and total pore volume of 0.27 cm^3^/g. The largest amount of nitrogen was adsorbed by the sample with KOH content of 50 wt.%. The specific surface area and TPV for this material equaled 665 m^2^/g and 0.36 cm^3^/g, respectively. Similarly, as with samples obtained using solid-based method, for the materials activated with KOH solution, a positive effect of the higher content of the activator agent on the value of the specific surface area and total pore volume was noticed. A slight increase in the S_BET_ and TPV for the material activated with KOH content of 10 wt.%, was noticed in comparison with the reference material, which equaled 496 m^2^/g and 0.27 cm^3^/g, respectively. However, in the case of the sample activated with KOH content of 70 wt.%, the specific surface area and total pore volume reached values of 918 m^2^/g and 0.49 cm^3^/g, respectively. Considering that the obtained results for both solid- and solution-based methods did not differ significantly, only one was used in further research.

It is worth noting that, during an attempt to change the activator agent to sodium hydroxide (N_50L_700_05) (Table 1, Figure 3), the amount of adsorbed nitrogen was lower than that of the sample activated with potassium hydroxide (K_50L_700_05), and the specific surface area of such a sample was equal to 557 m^2^/g while the total pore volume equaled 0.32 cm^3^/g. According to the low-temperature nitrogen adsorption-desorption studies, for the sample prepared excluding microwave treatment and activated with KOH content of 50 wt.% (K_50L_700_05_RFB) (Table 1, Figure 3), a slightly higher volume of adsorbed N_2_ was observed, in comparison with the same sample that was additionally treated using microwaves (K_50L_700_05). The specific surface area equaled 695 m^2^/g and 637 m^2^/g for the K_50L_700_05_RFB and K_50L_700_05 materials, respectively. In contrast to the above-mentioned materials, for the sample where potassium was not remove after the activation process (K 20L_700_05 WB) (Table 1, Figure 3), the surface area was very low, and equaled only 92 m^2^/g. The total pore volume was also very low and amounted to 0.07 cm^3^/g. This result shows that potassium can be present on the carbon spheres’ surface, blocking access to the pores and simultaneously preventing effective nitrogen adsorption.

The influence of the different temperatures and different activation times on the structural parameters of the obtained materials was also checked (Table 1, Figure 4). It can be clearly seen that a higher activation temperature resulted in an increase in specific surface area as well as total pore volume. The S_BET_ of the sample activated at 800 °C for 5 min was about 255 m^2^/g higher and a TPV of about 0.13 cm^3^/g was found compared to the sample activated at 700 °C under the same experimental conditions. An increase in the specific surface area, together with the increase in the carbonization temperature, was observed and described in our previous work [41]. A similar effect was noted when the activation time was extended from 5 to 60 min (Table 1, Figure 4) and the calculated specific surface area equaled 637 m^2^/g and 698 m^2^/g, but the increase was no longer significant.

A significant increase in specific surface area and TPV was observed for the sample with 70 wt.% KOH added after microwave treatment and before carbonization (Table 1, Figure 5). The specific surface area and total pore volume for K_70L_700_05_AR material equaled 734 m^2^/g and 0.48 cm^3^/g, respectively. These values were slightly lower than those achieved for the sample activated under the same experimental conditions but with the addition of KOH after the carbonization step. For K_70L_700_05, material specific surface area and total pore volume equaled 918 m^2^/g and 0.49 cm^3^/g, respectively. A similar dependence was observed for the material with 50 wt.% KOH. The specific surface area of the sample K_50L_700_05_AR was 558 m^2^/g, and was about 80 m^2^/g lower that of K_50L_700_05 (S_BET_ = 637 m^2^/g). Even lower values were observed for the material activated with 50 wt.% KOH, but added before the microwave reactor (Table 1, Figure 5). The specific surface area of K_50L_700_05_BR equaled only 450 m^2^/g. The material activated after treatment in the microwave reactor with 10 wt.% KOH (K_10L_700_05_AR) exhibited the lowest specific surface area of only 252 m^2^/g, which was much lower in comparison with the S_BET_ of the sample K_10L_700_05. This can indicate that the activation of pre-made spheres is slightly more effective than the activation of polymer spheres that were not yet subjected to the carbonization process.

As shown in Figure 2, Figure 3 and Figure 4, isotherms of type I according to the IUPAC classification were determined, indicating the microporous character of both the reference sample and the samples activated using solid- and solution-based methods. In the case of the materials, for which KOH was added before or after the microwave reactor (Figure 5), the nitrogen isotherms were of type II, which could suggest an increase in mesopore’s contribution to the materials. The obtained isotherms characterized the H4 type of hysteresis loop (Figure 5), which is often observed in micro-mesoporous carbon materials and can be found on ink-bottle type pores. The exception was the sample activated with the lowest KOH content of 10 wt.%, for which a type I isotherm was received, as in the materials shown in Figure 2, Figure 3 and Figure 4, which can indicate that the KOH content was too low and had no impact on the creation of more mesopores.

From the available papers, it is can be seen that using potassium as an activating agent, not only in the form of potassium hydroxide [42] but also, e.g., as potassium carbonate [31,42], contributes to the formation of a microporous structure. Due to the fact that, for effective CO_2_ adsorption under the atmospheric conditions of a pore content below 1 nm is crucial [36,43,44], the microporosity of the obtained materials was investigated in more depth. Figure 6 and Figure 7 show the PSD of materials activated with different KOH contents using solid-based and solution-based methods, respectively. The activation of carbon spheres with KOH changed the distribution of the ultramicropores regardless of the activation method and KOH content. Compared to the reference sample (RF), the development of 0.35 nm and 0.55 nm pores was observed, whereas the content of 0.85 nm pores was diminished. The overall content of ultramicropores, which was V_s_ < 1 nm for the samples activated with KOH content of 10 wt.% or 20 wt.%, regardless of the application method, was similar to the nonactivated reference sample (Table 1). Consequently, the amount of KOH was too low to achieve the efficient activation of carbon spheres. Nevertheless, as the amount of KOH was increased and reached 50 wt.%, the activation of carbon materials was noticed. Regardless of the KOH application method, samples activated with 50 wt.% KOH were associated with a higher Vs contribution, from 0.19 cm^3^/g for the reference material to 0.25 cm^3^/g for both K_50L_700_05 and K_50S_700_05 materials. Considering that only KOH content affected the activation process (not the application method), one additional sample was prepared using a solution method with KOH content of 70 wt.% and the calculated value of Vs for this material was the highest among all presented samples, equaling 0.31 cm^3^/g (K_70L_700_05).

The influence of the type of activating agent on the ultramicropores volume was determined, and a PSD comparison between the materials activated using potassium hydroxide and sodium hydroxide is presented in Figure 8. The sample activated using NaOH had less microporosity compared to the sample activated with KOH. These results are consistent with the data presented in the available papers and proved that carbon activated with potassium hydroxide had a more microporous structure than activated carbon produced using sodium hydroxide [45,46,47].

The influence of the temperature and time of the activation process on the ultramicropores volume was also investigated, and the PSD of the obtained samples is given in Figure 9. An increase in temperature from 700 °C to 750 °C, and even 800 °C, caused an increase in total pore volume in the following order: 0.35 cm^3^/g < 0.37 cm^3^/g < 0.48 cm^3^/g. Although the highest TPV was achieved at 800 °C, the increased contribution of ultramicropores V_s_ with dimensions smaller than 1.0 was not observed and the overall Vs content slightly increased, from 0.25 cm^3^/g for K_50L_700_05 to 0.27 cm^3^/g for K_50L_750_05 and 0.28 cm^3^/g for K_50L_800_05. Furthermore, extending the activation time from 5 min for sample K_ 50L_700_05 to 60 min for sample K_50L_700_60, keeping temperature constant, resulted in a slight development in V_s_ volume. It can be concluded that the proper choice of activation temperature more strongly affects this than the exposure length of the sample at that temperature.

The increase in KOH content added after microwave treatment from 10 wt.% to 50 wt.% and 70 wt.% resulted in an increase in ultramicropore volume from 0.12 cm^3^/g to 0.26 cm^3^/g and 0.32 cm^3^/g, respectively (Figure 10). Moreover, the increased contribution of pores of 0.35 nm in size can be noticed. For the sample where 50 wt.% KOH was added before microwave treatment, K_50L_700_05_BR, the contribution of pores of 0.35 nm is lower than for the sample where the same amount of KOH was added after microwave treatment, K_50L_700_05_AR.

### 2.3. CO_2_ Adsorption Properties

The influence of the activation process on CO_2_ uptake was studied and the values of CO_2_ adsorption capacity at 0 °C and 25 °C, calculated based on CO_2_ adsorption isotherms presented in Figure 11, Figure 12, Figure 13 and Figure 14, are listed in Table 2.

The carbon dioxide sorption isotherms of the materials activated using a KOH solid-based method are given in Figure 11. For the material with KOH content of 10 wt.%, no changes were observed in the CO_2_ sorption ability, and for the sample with KOH content of 20 wt.%, only a slight increase in CO_2_ adsorption was noticed in comparison with the reference sample RF. The highest CO_2_ adsorption value was obtained for the sample treated with KOH content of 50 wt.% and equaled 4.50 mmol/g at 0 °C.

The carbon dioxide sorption isotherms of the materials activated using a KOH solution-based method are presented in Figure 12. As for the samples obtained using KOH solid-based method, for the materials with KOH content of 10 wt.% and 20 wt.%, activated using solution-based method, only a slight increase in CO_2_ adsorption was noticed. An increase in the KOH content in the sample to 50 wt.% resulted in a significant increase in CO_2_ adsorption value, from 3.25 mmol/g for the reference material RF to 4.46 mmol/g for K_50L_700_05 sample. This value was similar to that obtained for the sample activated using the solid-based method with the same KOH content. Thus, KOH application method did not affect the CO_2_ adsorption ability. As mentioned in Section 2.2, regardless of the KOH application method, samples activated with a 50 wt.% KOH were associated with higher Vs contribution, which was reflected in the higher CO_2_ adsorption values. The highest CO_2_ adsorption values, 5.32 mmol/g, were expressed in the sample activated with KOH content of 70 wt.% (K_70L_700_05).

The adsorption capacity of carbon material activated using sodium hydroxide was also checked (Figure 12). The sample was treated using a solution-based method with KOH content of 50 wt.% expressed a much higher CO_2_ uptake ability, at 0 °C (4.46 mmol/g), compared to the same sample, activated using NaOH (3.82 mmol/g). This was connected with the lower volume of micropores for the sample activated using NaOH compared to KOH. It is worth noting that, on the sample synthesized without microwave treatment (K_50L_700_05_RFB, Figure 12), a slightly higher amount of CO_2_ was adsorbed than in the sample obtained using a microwave reactor (K_50L_700_05). In the first case, 4.91 mmol/g was noted, while 4.46 mmol/g (at 0 °C) was obtained in the second. This is consistent with the surface area and TPV. As expected for the sample in which potassium was not removed after the activation process (K_20L_700_05_WB), the lowest CO_2_ adsorption value was achieved, which equaled only 1.65 mmol/g and 1.29 mmol/g at 0 °C and 25 °C, respectively.

Comparing the sample obtained using the solution-based method with KOH content of 50 wt.%, activated at 700 for 5 min, with the samples obtained under the same experimental conditions but treated in the high-temperature furnace at 750 °C and 800 °C (Figure 13), it was found that the increase in temperature had practically no effect on the CO_2_ adsorption ability of carbon material. An increase in the activation temperature from 700 °C to 750 °C only slightly improved the CO_2_ adsorption values, from 4.46 mmol/g for the K_50L_700_05 to 4.93 mmol/g and 4.93 mmol/g for K_50L_750_05 and K_50L_800_05, respectively. It is worth noting that, despite the same CO_2_ adsorption values being used for samples K_50L_750_05 and K_50L_800_05, the differences in the surface area and total pore volume were quite significant. The volume of micropores < 2 nm was also greater for the sample carbonized at 800 °C. Only the volume of ultramicropores hardly differed, confirming that ultramicropores are mainly responsible for the adsorption of CO_2_. Treating the samples at 700 °C for 60 min, instead of 5 min, did not significantly improve the CO_2_ adsorption at 0 °C, but the CO_2_ adsorption value at 25 °C increased from 2.96 mmol/g for sample K_50L_700_05 to 3.50 mmol/g for sample K_50L_700_60.

To investigate the prepared material’s susceptibility to activation at different preparation steps, the adsorption properties of the materials prepared where KOH was added before (BR) or after microwave treatment (AR), but before the carbonization step, were compared. CO_2_ isotherms are presented in Figure 14. The addition of 10 wt.% liquid KOH after microwave treatment had almost no effect on CO_2_ adsorption ability at 0 °C, and the calculated values equaled 3.25 mmol/g for RF material and 3.15 mmol/g for the K_10L_700_05_AR sample. Nevertheless, a further increase in the concentration of the activating agent of up to 50 wt.% (the sample K_50L_700_05_AR) resulted in higher CO_2_ uptake values of 4.62 mmol/g. The sample obtained under the same experimental conditions, but for which only the activating agent was added before (not after) microwave treatment (K_50L_700_05_BR), exhibited a lower CO_2_ adsorption capacity of 3.89 mmol/g. The highest CO_2_ adsorption (5.09 mmol/g) value was noticed for the material activated after microwave treatment with 70 wt.% KOH (K_70L_700_05_AR). Nevertheless, this value was slightly lower than it was for the corresponding material activated after carbonization, K_70L_700_05, and expressed a higher CO_2_ adsorption ability of 5.32 mmol/g at 0 °C. The increase in KOH content added after microwave treatment from 10 wt.% to 50 wt.% and 70 wt.% resulted in an increase in ultramicropore volume from 0.12 cm^3^/g to 0.26 cm^3^/g and 0.32 cm^3^/g, respectively (see Section 2.2, Table 1, Figure 10). Moreover, the consecutively higher contribution of pores of 0.35 nm in size could be noticed. These results confirm that the presence of the ultramicropores below 0.4 nm is a very important factor, which affects the effective CO_2_ adsorption ability of carbon material under ambient conditions [48]. For the sample where 50 wt.% KOH was added before microwave treatment, K_50L_700_05_BR, the contribution of pores of 0.35 nm is lower than for the sample where the same amount of KOH was added after microwave treatment, K_50L_700_05_AR, and the higher contribution of 0.8 nm pores did not cause a substantial increase in the CO_2_ adsorption ability of this sample.

It was found that the CO_2_ adsorption isotherms presented in Figure 11, Figure 12, Figure 13 and Figure 14 can be described by the equation introduced by Sips [49], often called the Freundlich-Langmuir equation:(1)$$a=a_{m}\cdot \frac{K\cdot p^{n}}{1+K\cdot p^{n}}$$
where: a—adsorption (concentration in the solid phase) [mmol/g]; a_m_—maximum adsorption (initial concentration of active sites in the solid phase) [mmol/g]; K—equilibrium constant [bar^−n^]; n—constant; p—adsorbate pressure [bar].

An exemplary fit for CO_2_ isotherms obtained at 0 °C and 25 °C for sample K_70L_700_05 is presented in Figure 15. Equation (1) can be justified by assuming that adsorption can be written by the equation:(2)$$A+\mathrm{nB}\to {\mathrm{AB}}_{n}$$
where: A—active sites of the adsorbent; B—the adsorbate.

Taking the equilibrium state according to the law of action mass:(3)$$K=\frac{a}{\left(a_{m}-a\right)\cdot p^{n}}$$
from which Equation (1) follows.

Based on the available literature, it can be assumed that the constant n is a dimensionless parameter that qualitatively characterizes the heterogeneity of the adsorbate-adsorbent system [50]. Therefore, the Sips isotherm shall only be used to describe monolayer adsorption systems. Multilayer adsorption may occur at high pressures for the systems being studied. Therefore, the Sips model can be used to describe the adsorption at the pressure range of interest, and the am parameter should not be considered the ultimate saturation adsorption. The parameters calculated for Equation (1) are summarized in Table 2.

The values of the K constant are lower at higher temperatures, which clearly indicates the exothermic nature of the process. The dependence of the equilibrium constant on temperature is given by equation:(4)$$K=K_{0}\exp\left(\frac{-\Delta H}{R\cdot T}\right)$$
where: ΔH—average heat of adsorption [J/mol]; R—gas constant [J/molK], T—temperature [K]; K_0_—entropy coefficient.

The average adsorption heat, determined based on the Equation (4), is also summarized in Table 2. The K_70L_700_05, K_50L_700_05_RFB and K_70L_700_05_AR samples characterize the lowest CO_2_ adsorption heat, which indicates that the CO_2_ molecules mainly interact with the carbon surface due to weak dispersion forces. This could be a result of the high degree of graphitization, which also results in relatively high am values for these samples. The high adsorption energy for K_20L_700_05_WB material indicates chemisorption’s significant contribution to the CO_2_ sorption of this material. Chemisorption can be associated with unwashed KOH, which can react with CO_2_ molecules to form potassium carbonate. The almost two-fold increase in the am value at 25 °C in comparison to 0 °C also confirms this explanation because it is known that the higher temperature can lead to the participation of chemisorption. For all samples, the value of parameter n indicates that they are characterized by their significant surface heterogeneity.

In the case of samples K_10L_700_05_AR, and K_50L_700_05_BR, the constant n is close to 1, indicating that the adsorption process proceeds according to the Langmuir theory. This suggests that these adsorbents have an energetically homogeneous surface and a constant adsorption energy. This could also be the result of the strong graphitization of the surface with a small proportion of surface functional groups.

The above results show that the best parameters were calculated in the case of CO_2_ adsorption for the sample K_70L_700_05, K_50L_700_05_RFB and K_70L_700_05_AR. Due to the relatively high am values and the low adsorption heat, these materials should demonstrate good efficiency in CO_2_ adsorption and, due to the relatively low heat of adsorption, they should be easy to regenerate.

## 3. Materials and Methods

### 3.1. Preparation of Materials

Carbon spheres obtained from resorcinol and formaldehyde were used as a reference material. For this purpose, 2.4 g of resorcinol was dissolved in an aqueous alcohol solution composed of 240 mL distilled water and 96 mL ethanol. To adjust pH to ~9, ammonium hydroxide (25 wt.%) was slowly dropped into the beaker. Then, 3.6 mL of formaldehyde (37 wt.%) was added and the whole was mixed using a magnetic stirrer at ambient conditions to facilitate polycondensation reaction. After 24 h, the mixture was transferred into a microwave reactor (Ertec, Wroclaw, Poland) and treatment was carried out for 15 min under a reaction pressure of 20 at. The effect of microwave-assisted solvothermal process parameters on the carbon dioxide adsorption properties of microporous carbon spheres was studied and presented in more depth in [40]. Next, the products were dried for 24 h at 80 °C, and then carbonized in a high-temperature furnace (HST 12/400 Carbolite) (Carbolite, Derbyshire, UK) under argon atmosphere with the temperature increasing from 20 °C to 350 °C at a heating rate of 1 °C/min and holding time of 2 h, and from 350 °C to 700 °C at a heating rate of 1 °C/min. After 2 h, the sample was cooled to room temperature under argon atmosphere. The final product was washed with distilled water and dried for 48h at 80 °C in air.

Activated carbon spheres were synthesized in two ways: a solid-based and solution-based method. In the case of the solid-based method, the appropriate amount of potassium hydroxide was ground in an agate mortar and mechanically mixed with 1 g of carbon spheres. Next, the samples were treated in a high-temperature furnace (HST 12/400 Carbolite) (Carbolite, Derbyshire, UK) under an argon atmosphere with the temperature increasing from 20 °C to 700 °C at a heating rate of 10 °C/min. After 5 min, the samples were cooled to room temperature under argon atmosphere. To remove potassium from the material, the final product was boiled in distilled water under reflux for 1 h, and the pH of the filtrate was measured. The boiling procedure was repeated until a neutral pH was reached. Finally, the sample was dried for 48 h at 100 °C. Materials with an initial KOH content of 10 wt.%, 20 wt.%, and 50 wt.% were obtained and marked as: K_10S_700_05, K_20S_700_05, and K_50S_700_05, respectively.

In the second case, 1 g of carbon spheres, prepared as described above, was placed in a quartz boat and the appropriate amount of potassium hydroxide solution with a concentration of 5 mol/dm^3^ was added. To maintain the same total volume of impregnation agent, equal to 5 mL, an appropriate amount of distilled water was added to the quartz boat. The resulting solution was mixed for 1 h and dried at 100 °C for 24 h. Next, the sample was activated in a high-temperature furnace (HST 12/400 Carbolite) (Carbolite, Derbyshire, UK) under an argon atmosphere, with the temperature increasing from 20 °C to 700 °C at a heating rate of 10 °C/min; after 5 min, it was cooled to room temperature under argon atmosphere. To remove potassium from the sample, the final product was boiled in distilled water under reflux for 1 h and the pH of the filtrate was checked. The boiling procedure was repeated until a neutral pH was reached. Finally, the sample was dried for 48 h at 100 °C. The materials with an initial KOH content of 10 wt.%, 20 wt.%, 50 wt.% and 70 wt.% were obtained and marked as K_10L_700_05, K_20L_700_05, K_50L_700_05 and K_70L_700_05, respectively. Additionally, the material activated with sodium (instead of potassium) hydroxide content of 50 wt.%, was prepared in the same way and marked as Na_50L_700_05.

To evaluate the influence of potassium hydroxide on the adsorption properties of carbon spheres, the sample with KOH content of 20 wt.% was synthesized and, after activation, the material was not boiled in the distilled water under reflux, but was directly tested. The obtained sample was marked as K_20L_700_05_WB.

To determine the effect of activation temperature and activation time on the adsorption properties, the materials with KOH content of 50 wt.% were obtained according to the same procedure as described above, with the difference that activation proceeded at 750 °C (K_50L_750_05) or 800 °C (K _50L_800_05), or activation at 700 °C lasted not 5 min but 60 min (K_50L_700_60).

The influence of microwaves on the adsorption properties of the obtained materials was also checked. For these studies, carbon spheres were prepared in the same manner as the reference material, excluding the treatment in the microwave reactor. The obtained sample was activated using a solution-based method with KOH content of 50 wt.% at 700 °C for 5 min. The other synthesis steps remained unchanged, and the sample was marked as K_50L_700_05_RFB.

In the above-mentioned procedures, potassium hydroxide was added to the previously prepared carbon spheres. For comparison purposes, potassium hydroxide was also introduced during the synthesis of carbon spheres, that is, 50 wt.% KOH was added before microwave treatment (K_50L_700_05_BR) or 10 wt.% (K_10L_700_05_AR), 50 wt.% (K_50L_700_05_AR) or 70 wt.% KOH (K_70L_700_05_AR) was added after microwave treatment but before the carbonization step. Other experimental conditions were maintained as described above.

### 3.2. Characterization of the Materials

Morphology of the samples was investigated using SU8020 Ultra-High Resolution Field Emission Scanning Electron Microscope (Hitachi Ltd., Japan). Surface properties were determined using N_2_ adsorption/desorption isotherms performed on a QUADRASORB evoTM Gas Sorption automatic system (Quantachrome Instruments, Boynton Beach, FL, USA) at –196 °C. Before each adsorption experiment, samples were outgassed at 250 °C under a vacuum of 1 × 10^−5^ mbar for 12 h using a MasterPrep multi-zone flow/vacuum degasser from Quantachrome Instruments to remove adsorbed species that could intervene in the adsorption processes. The surface area (S_BET_) was determined in the relative pressure range of 0.05–0.3 and calculated based on Brunauer-Emmett-Teller (BET) equation. The total pore volume (TPV), was calculated from the volume of nitrogen held at the highest relative pressure (p/p0 = 0.99). The volume of micropores V_m_ with dimensions smaller than 2 nm was calculated as a result of integrating the pore volume distribution function using the DFT method; the volume of mesopores V_meso_ with dimensions from 2 to 50 nm was calculated from the difference in the total pore volume TPV and the volume of micropores V_m_. From CO_2_ adsorption isotherms, obtained using a QUADRASORB evoTM Gas Sorption automatic system (Quantachrome Instruments, Boynton Beach, FL, USA) at 0 °C, pore size distribution (PSD) and the volume of ultramicropores V_s_ with dimensions smaller than 1.0 nm (<1 nm) was determined and calculated by integrating the pore volume distribution function using the NLDFT method.

### 3.3. CO_2_ Adsorption Experiments

Carbon dioxide adsorption isotherms at 0 °C and 25 °C were measured using the QUADRASORB evoTM Gas Sorption automatic system (Quantachrome Instruments, Boynton Beach, FL, USA), as mentioned above, in the pressure range between 0.01 and 0.98 bar.

## 4. Conclusions

As expected, the activation of carbon spheres using potassium hydroxide resulted in an increase in specific surface area compared with the non-activated material. The values of specific surface area correlated with the values of the total pore volume. It should also be noted that higher surface area and, therefore, higher total pore volume did not always translate into higher CO_2_ adsorption values. The size of the micropores plays an important role in this process. It was found that the use of solid or liquid potassium hydroxide to activate carbon spheres has practically no effect on CO_2_ adsorption. Potassium hydroxide was a more efficient activator than sodium hydroxide. The omission of microwave treatment step during the preparation of carbon spheres had no effect on the CO_2_ adsorption values. The best adsorption results were observed for the samples activated after the carbonization step, using 70 wt.% KOH, for which the value of CO_2_ adsorption reached 5.32 mmol/g at 0 °C.

## Figures and Tables

**Figure 1 molecules-27-05379-f001:**
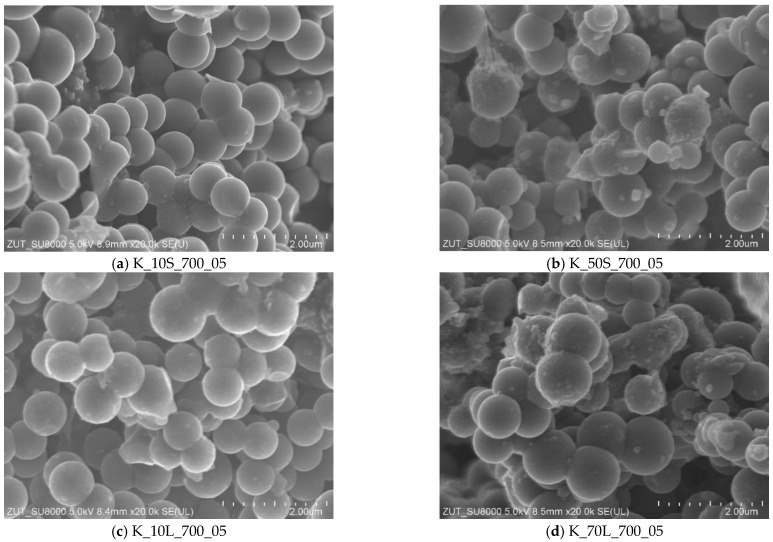
SEM images of the samples activated using a solid-based method with KOH content of 10 wt.% (**a**) and 50 wt.% (**b**) or a solution-based method with KOH content of 10 wt.% (**c**) and 70 wt.% (**d**).

**Figure 2 molecules-27-05379-f002:**
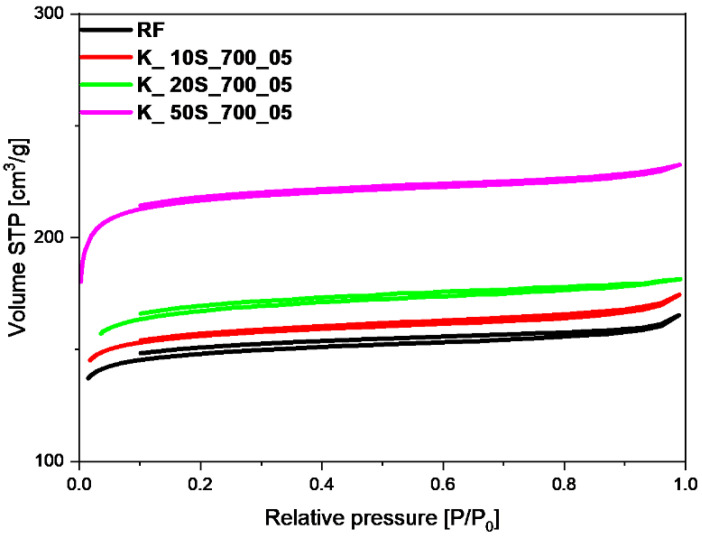
Nitrogen adsorption-desorption isotherms of reference material (RF) and samples activated using solid-based method with different KOH content.

**Figure 3 molecules-27-05379-f003:**
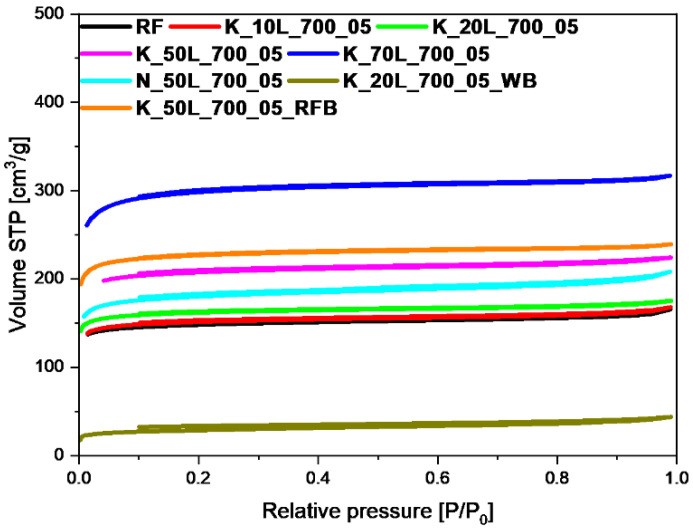
Nitrogen adsorption-desorption isotherms of reference material (RF), and samples activated using solution-based method with different KOH content.

**Figure 4 molecules-27-05379-f004:**
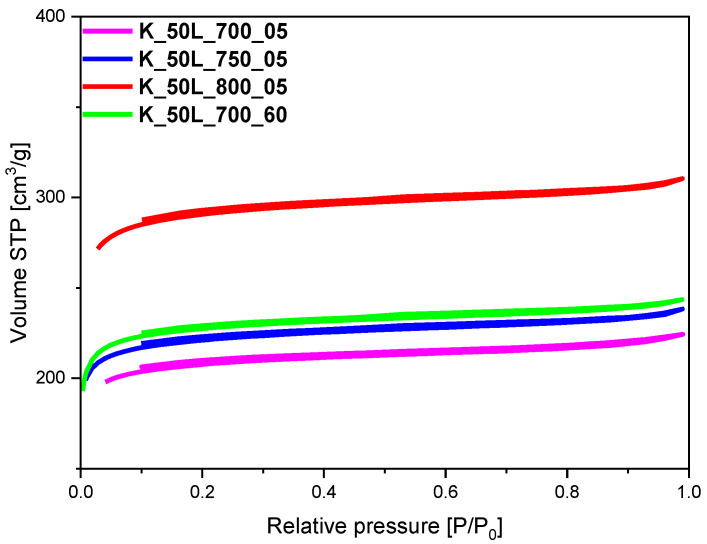
Nitrogen adsorption-desorption isotherms of the samples activated at different temperatures and with different dwell times.

**Figure 5 molecules-27-05379-f005:**
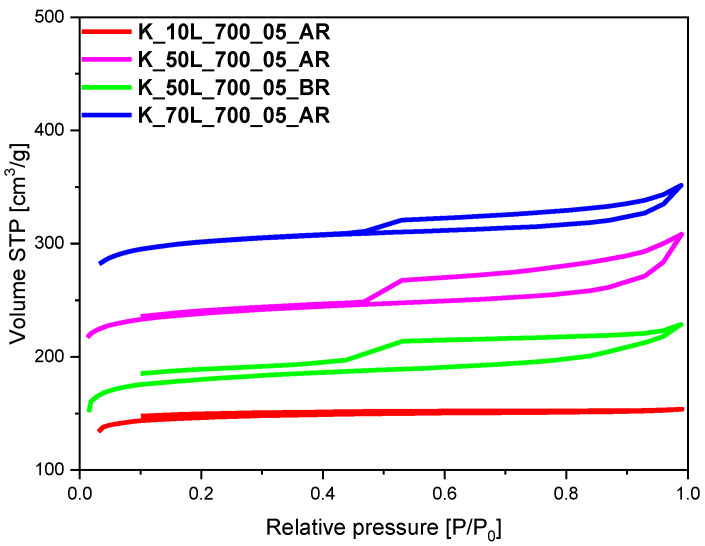
Nitrogen adsorption-desorption isotherms of the sample with KOH added before (K_10L_700_05_BR) or after (K_10L_700_05_AR; K_50L_700_05_AR; K_70L_700_05_AR) microwave reactor.

**Figure 6 molecules-27-05379-f006:**
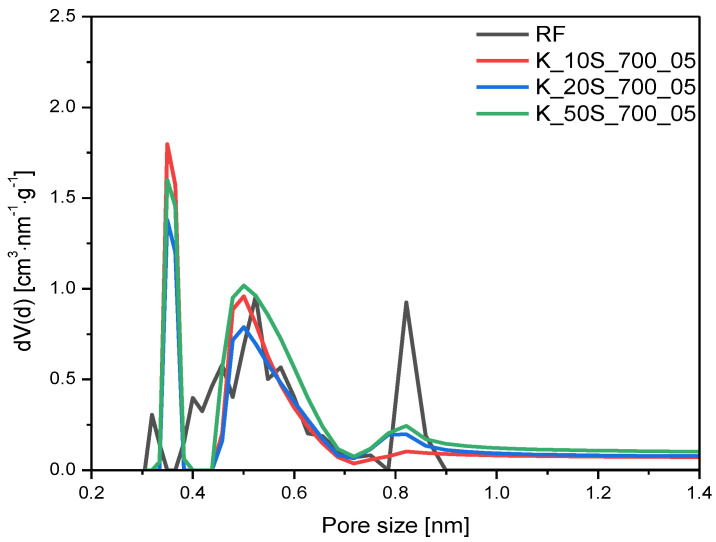
PSD of the samples activated with different content of KOH using a solid-based method.

**Figure 7 molecules-27-05379-f007:**
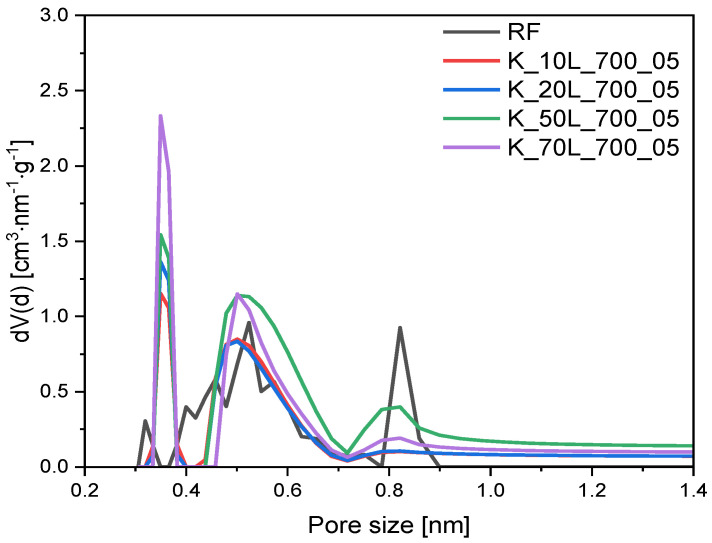
PSD of the samples activated with different content of KOH using solution-based method.

**Figure 8 molecules-27-05379-f008:**
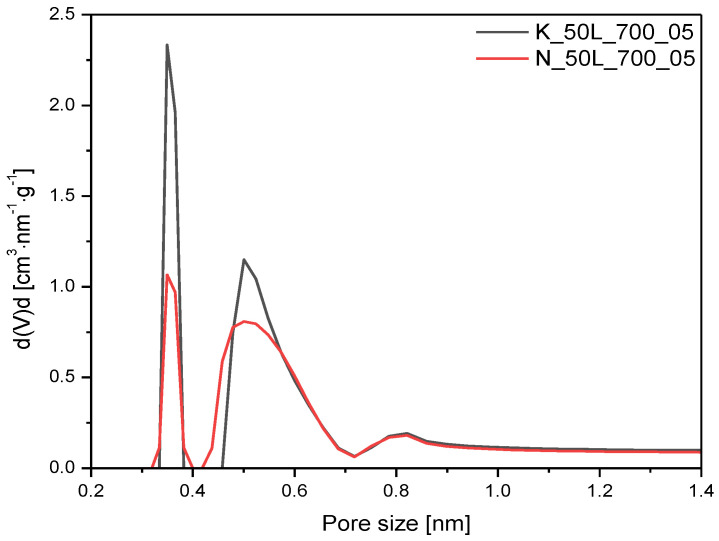
PSD of the samples activated using different chemical activator.

**Figure 9 molecules-27-05379-f009:**
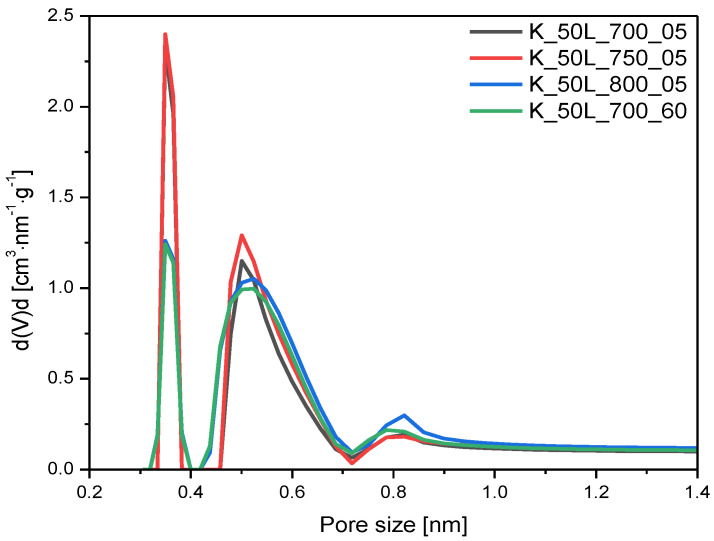
PSD of the samples activated at different activation temperatures and with different dwell times.

**Figure 10 molecules-27-05379-f010:**
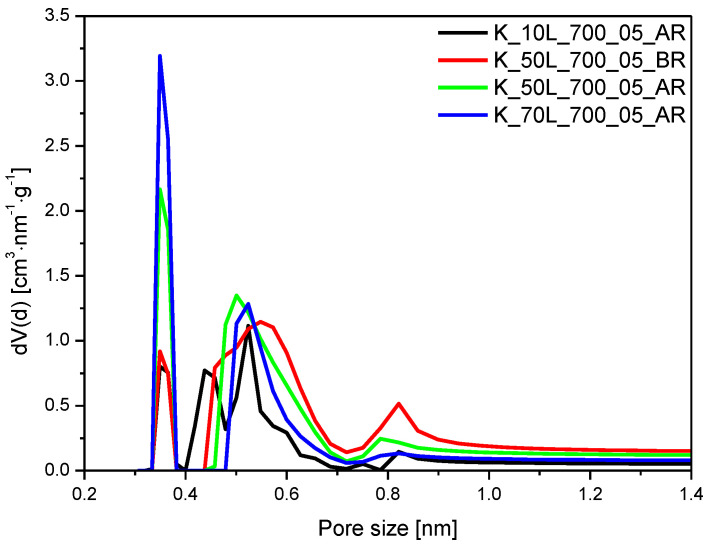
PSD of the samples activated with KOH and added at different preparation steps.

**Figure 11 molecules-27-05379-f011:**
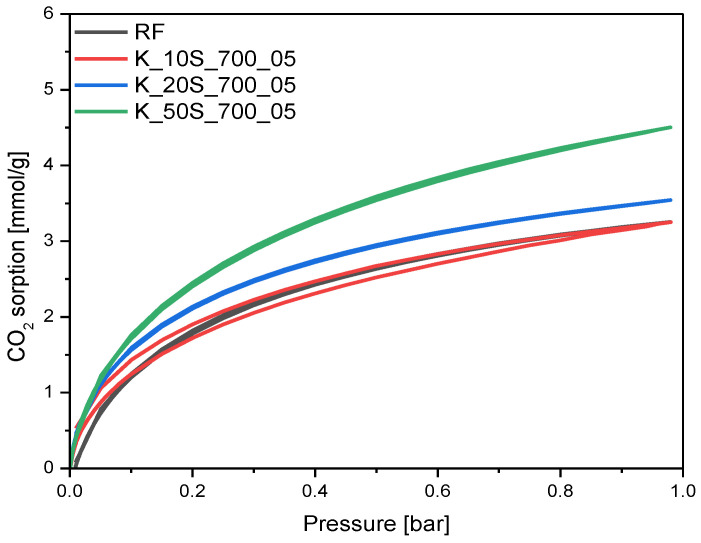
CO_2_ sorption isotherms at 0 °C of the samples activated using solid-based method with different KOH content.

**Figure 12 molecules-27-05379-f012:**
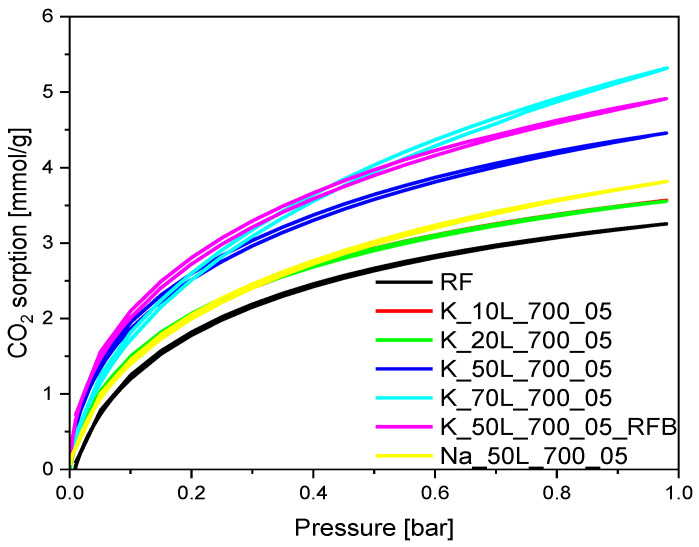
CO_2_ sorption isotherms at 0 °C of the samples activated using solution-based method with different KOH content and using different chemical activators.

**Figure 13 molecules-27-05379-f013:**
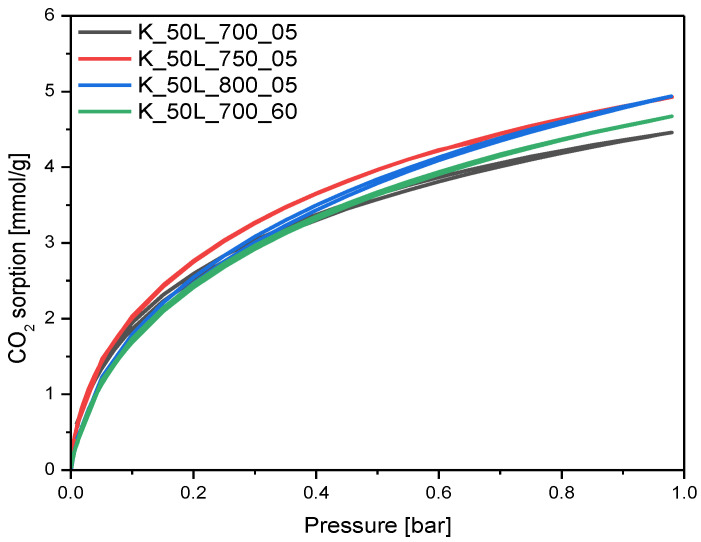
CO_2_ sorption isotherms at 0 °C of the samples activated at different temperatures and with different dwell times.

**Figure 14 molecules-27-05379-f014:**
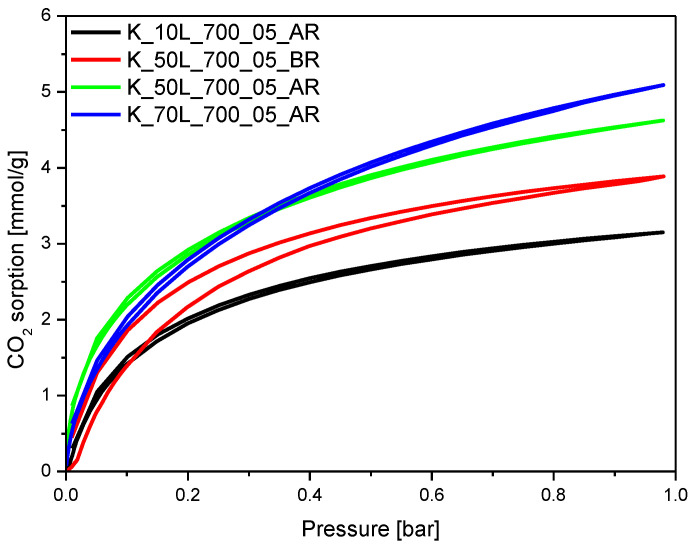
CO_2_ sorption isotherms at 0 °C of the samples for which KOH was added at different preparation steps.

**Figure 15 molecules-27-05379-f015:**
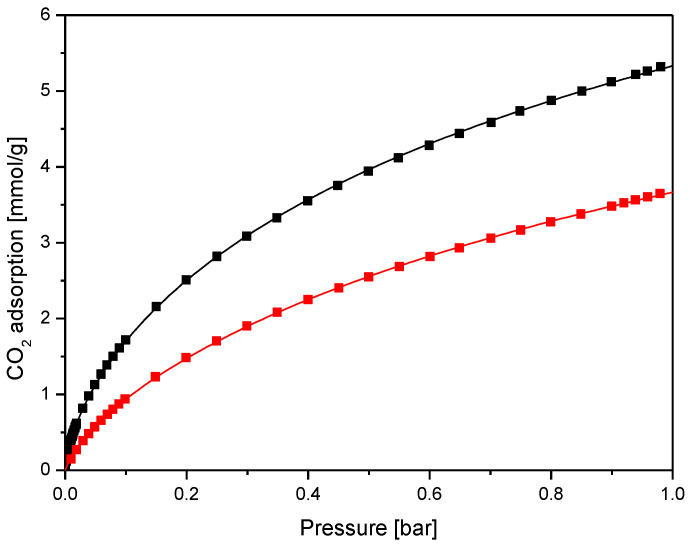
Experimental isotherms for the sample K_70L_700_05 at 0 °C (black) and 25 °C (red) and their corresponding isotherms, calculated from Freundlich-Langmuir equation.

**Table 1 molecules-27-05379-t001:** Structural parameters of the materials studied.

Sample Name	S_BET_	TPV	V_s_ (<1 nm)	V_m_ (<2 nm)	V_meso_
	[m^2^/g]	[cm^3^/g]	[cm^3^/g]	[cm^3^/g]	[cm^3^/g]
RF	455	0.26	0.19	0.22	0.04
K_10S_700_05	480	0.27	0.19	0.22	0.05
K_20S_700_05	515	0.28	0.18	0.24	0.04
K_50S_700_05	665	0.36	0.25	0.31	0.05
K_10L_700_05	496	0.27	0.20	0.23	0.04
K_20L_700_05	465	0.26	0.19	0.21	0.05
K_50L_700_05	637	0.35	0.25	0.30	0.05
K_70L_700_05	918	0.49	0.31	0.42	0.07
Na_50L_700_05	557	0.32	0.22	0.26	0.06
K_20L_700_05_WB	92	0.07	0.01	0.04	0.03
K_50L_750_05	679	0.37	0.27	0.32	0.05
K_50L_800_05	892	0.48	0.28	0.42	0.06
K_50L_700_60	698	0.38	0.27	0.32	0.06
K_50L_700_05_RFB	695	0.37	0.27	0.32	0.05
K_10L_700_05_AR	252	0.14	0.12	0.12	0.02
K_50L_700_05_BR	450	0.29	0.21	0.21	0.08
K_50L_700_05_AR	558	0.35	0.24	0.26	0.09
K_70L_700_05_AR	734	0.48	0.29	0.32	0.16

S_BET_—specific surface area; TPV—total pore volume; V_s_—the volume of ultramicropores with diameter smaller than 1nm; V_m_—the volume of micropores with diameter smaller than 2nm; V_meso_—the volume of mesopores with diameter from 2 to 50 nm.

**Table 2 molecules-27-05379-t002:** Gas adsorption parameters determined based on Equation (1) and average adsorption heat determined based on Equation (4).

	Temperature 0 °C				Temperature 25 °C				
Sample Name	CO_2_	a_m_	K	n	CO_2_	a_m_	K	n	−ΔH
-	mmol/g	mmol/g	bar^−n^	-	mmol/g	mmol/g	bar^−n^	-	J/mol
RF	3.25	-	-	-	2.43	-	-	-	-
K_10S_700_05	3.25	5.45	1.86	0.667	2.11	4.95	1.13	0.731	13483
K_20S_700_05	3.54	6.64	0.957	0.625	2.41	6.14	0.602	0.699	12541
K_50S_700_05	4.50	7.87	1.33	0.688	3.12	8.77	0.639	0.676	19832
K_10L_700_05	3.57	5.24	2.13	0.75	2.43	5.02	1.13	0.773	17150
K_20L_700_05	3.55	5.28	2.05	0.727	2.50	5.32	1.05	0.732	18101
K_50L_700_05	4.46	8.85	1.02	0.586	2.96	8.14	0.66	0.642	11778
K_70L_700_05	5.32	12.80	0.697	0.665	3.35	10.45	0.54	0.739	6905
Na_50L_700_05	3.82	6.52	1.41	0.724	2.64	7.22	0.667	0.683	20252
K_20L_700_05_WB	1.65	2.86	1.31	0.696	1.29	4.78	0.414	0.599	31165
K_50L_750_05	4.93	9.31	1.13	0.616	3.56	8.26	0.765	0.689	10554
K_50L_800_05	4.93	9.26	1.15	0.713	3.64	11.4	0.469	0.683	24266
K_50L_700_60	4.67	8.11	1.36	0.725	3.50	9.13	0.623	0.71	21122
K_50L_700_05_RFB	4.91	11.06	0.802	0.557	3.21	9.14	0.624	0.659	6790
K_10L_700_05_AR	3.15	3.73	4.76	0.926	2.71	3.39	2.77	0.964	14648
K_50L_700_05_BR	3.89	4.95	3.67	1	2.77	3.71	2.77	1	7612
K_50L_700_05_AR	4.62	8.57	1.185	0.536	3.16	6.76	0.886	0.628	7867
K_70L_700_05_AR	5.09	10.49	0.954	0.627	3.30	7.77	0.746	0.728	6654

## Data Availability

The data presented in this study will be available upon request.

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
