# Peer review of "CO2 Adsorption Study of Potassium-Based Activation of Carbon Spheres"

_molecules, 2022, doi:10.3390/molecules27175379_

Round 1

Reviewer 1 Report

In this manuscript, microporous spherical carbon materials were prepared and the effect of various activation parameters such as activation temperature/time and chemical agent content were studied. The CO2 adsorption capacities of the prepared materials were also evaluated. The manuscript provided some useful information. However, many questions need to be addressed before the manuscript can be accepted for publication.

1)    Chemical activation and thermal activation are standard methods for carbon materials. The authors need to address what is the novelty of this manuscript.

2)    The manuscript is a pile of observations of the experiments but lacks some scientific explanation of the results.

3)    It would be better if the author could address the microwave treatment's purpose and necessity in the synthesis process.

4)    Figure 4, what’s the cause of the generation of mesopores? Why would the KOH concentration matter in terms of the generation of mesopores?

5)    Page 8, the authors should provide data about the “lower mechanical resistance” of K_50L_700_05_RFB.

6)    Figure 6, adsorption isotherms of K_10L_700_05 was not shown in the figure.

7)    Figure 10-14, it would be better if the authors could compare the PSD calculated using N2 isotherms and CO2 isotherms.

8)    Tables shouldn’t be across pages.

Reviewer 2 Report

The manuscript presents some good material about the variation in the production of carbon spheres. However, there are not enough characterizations that help to discuss the mechanisms involved in the potassium adsorption processes. I strongly recommend that the SEM, TGA and FTIR analyzes are included (before and after adsorption), in this way it will be possible to deepen the proposal of this manuscript.

Author Response

Dear Reviewer,

we are very grateful to the Reviewer for time and effort devoted to our manuscript. The Reviewer’s comments enabled us to significantly improve the manuscript. The detailed answers are below.

According to your advice, we added some examples of SEM images, but we decided not to add the results of thermogravimetric and FTIR analysis, as they did not bring valuable information.

This work focuses on the adsorption of CO2 on potassium hydroxide activated carbon spheres, not on the potassium adsorption mechanism. In the manuscript we investigated the following impacts: two methods of activation, activator concentrations, type of activator, activation temperature, use of microwaves, removing of activator. To underline it, we changed the title adding "CO2 adsorption study of ..."

It was not the purpose of this work to clarify "the mechanisms involved in the potassium adsorption processes" because potassium hydroxide was only used as an activator agent and residual potassium - as described in section 3.1 „In order to remove potassium from the material, the final product was boiled in distilled water under reflux for 1 h, and the pH of the filtrate measured. The boiling procedure was repeated until a neutral pH was reached.”- was removed after activation process. The use of potassium hydroxide only developed the specific surface and porous structure of the obtained materials, but not introduced potassium to carbon material. The main effect of the activation deals with an increase of microporosity, which has been shown in detail.

Of course, you are right that the use of potassium hydroxide can introduce functional oxygen groups on the surface of the carbon spheres, but they are not responsible for the CO2 adsorption process. In our opinion, the addition of data from thermogravimetric or FTIR analysis will unnecessarily extend the length of the paper, not bringing significant contribution in discussing the process of CO2 adsorption.

Nevertheless, thank you for this suggestion. We are also investigate the behavior of these carbon materials in the processes of adsorption of pollutants from the liquid phase and in this case - depending on the nature of the impurities -  the amount and type of functional groups may be significant and can have the influence on the adsorption process. Therefore, in the work on the adsorption studies from the liquid phase, we will include the results of both thermogravimetric and FTIR analysis.

Reviewer 3 Report

The study of activation techniques for carbon materials to enhance their CO2 adsorption characteristics is the focus of this research. Despite the appropriate experimental design, the data are presented in an unclear manner. This is a result of the paper's poorly thought-out structure. I advise an extremely substantial revision. The paper contains a lot of experimental material and there is a high probability of preparing a good publication.

The list of key notes:

1.         Why are the synthesized carbon materials called carbon spheres? The shape of the particles is not discussed in the paper.

2.         Nitrogen adsorption-desorption isotherms (Figs.1-4) are not informative. I recommend moving them in the Support Information. The Table 1 (Section 2.1. “Surface area and total pore volume analysis”) provides the necessary information. Unfortunately, the authors did not indicate in the Section 3.2 the method of calculation of Vs (the volume of ultramicropores).

3.         I don't see why, after the discussion of the CO2 adsorption results (Section 2.2) the data on the pore size distribution for the synthesized carbon materials was discussed again (Figs. 10-14) in Section 2.3 “Pore size distribution studies”. Their place is in Section 2.1. When I tried to understand the adsorption results of Section 2.2, I always referred to Table 1.

4.         In my opinion, the presentation of adsorption data is the main problem of this paper. Be sure to add a Table that shows the obtained values of adsorption at 0 and 25 °C for all samples (as Table 1). When reading the text of Section 2.2 it is not always clear which adsorption temperature is being discussed, the table will make the perception of the results clearer.

5.         Is the Freundlich-Langmuir isotherm (1) the most appropriate model describing the experimental data of this study? The dimensions of the equilibrium adsorption constant and other used parameters should be indicated hereinafter (required in the Table 2).

6.         The caption of the Fig.9 should be corrected. Experimental and modeling curve should be pointed out.

7.         The dependence of the equilibrium constant on temperature (Eq. (4)) is not corrected. Please see adsorption theory. It is well known that Gibbs’ free energy, entropy, and enthalpy (thermodynamic parameters) may be calculated from the equilibrium constant on temperature.

8.         Usually, the equation like Eq. (4) with a minus at E is used as an Arrhenius equation. E is the activation energy of the adsorption process (kinetic parameter). Usually this is a positive value in kJ/mol. The enthalpy or heat of adsorption is usually a negative value. What did the authors mean? It is necessary to clarify the equation (4), clearly show which energy parameter is estimated, check its sign and dimension in the Table 2.

9.         Note that measuring the adsorption at two temperatures is not enough to calculate both kinetic and thermodynamic parameters. Two points are not enough for a straight-line approximation. The plot of the approximation of experimental points and the confidence interval of errors must be presented when discussing the such results.

10.     To understand which characteristic of a carbon material (specific surface area, total pore volume, mesopore or micropore volume, pores radius and etc.) are important for CO2 adsorption, I recommend the authors to draw simple dependences, where the y axis shows the adsorption value, and the x axis shows the parameter of the porous structure. To present such results, the authors have an obvious advantage - many measurements. This will allow more accurate conclusions to be done.

11.     The text of the paper should be carefully rewritten. I give only one remark as an example. As for me, it is not obvious from Table 1 that the solid phase activation method is worse than via the liquid phase. But the authors wrote that that the liquid-phase activation method is more preferable. What are parameters analyzed? And there are many such examples. I encourage authors to make their results understandable to readers.

Round 2

Reviewer 1 Report

The authors have revised the manuscript accordingly. It would be better if the authors could include some information about microwave heating in the introduction.

Author Response

Dear Reviewer,

We included some information about microwave heating in the Introduction part (pink highlighting of the text). Additionally, in the Experimental part we indicated that: The effect of microwave assisted solvothermal process parameters on carbon dioxide adsorption properties of microporous carbon spheres was studied and presented more in depth in work [40].

Reviewer 2 Report

The authors made an improvement  in the manuscript during the review process. Therefore, I recommend this work for publication.

Author Response

Dear Reviewer,
thank you once again for your time and effort devoted to our manuscript. 

Reviewer 3 Report

The authors tryed to address all my questions and revised the text.

But, I am not satisfied with the presentation of the equation (4) in the revised text. If its basis is the Van't Hoff equation, then it is required to present the relationship between the parameters of the Van't Hoff equation and and the "average adsorption energy". It is well known that the Van't Hoff equation is traditionally used to  calculate the enthalpy and entropy of the adsorption process. The term "average adsorption energy" is usually introduced for the Dubin-Radushkevich isotherm.

Some design problems  may be corrected at the subsequent stage of paper publishing, .

Author Response

Dear Reviewer,

thank you once again for your time and effort devoted to our manuscript.

As suggested, we changed the notations in Equation (4) (pink highlighting of the text).